# *Legionella* in Hot Water Heat Pump (HWHP) Systems

**DOI:** 10.3390/microorganisms13051134

**Published:** 2025-05-15

**Authors:** Jodi Brookes, Helena Senior, Rebecca J. Gosling, Duncan Smith, Margaret Wade

**Affiliations:** 1Health and Safety Executive, Science and Research Centre, Harpur Hill, Buxton SK17 9JN, UK; 2Health and Safety Executive, Alnwick House, Benton Park View, Newcastle upon Tyne NE98 1YX, UK; 3Health and Safety Executive, Redgrave Court, Merton Road, Bootle L20 7HS, UK

**Keywords:** *Legionella*, hot water heat pumps, biofilms, risk of exposure

## Abstract

It is anticipated that by 2028 there will be a significant increase in the use of HWHP systems in Great Britain (GB). Such systems are considered a better, energy-efficient alternative to fossil fuel-based burners and furnaces, as they use electricity. There are concerns that these systems are susceptible to microbial contamination because they hold water at lower temperatures. In particular, the concern is regarding *Legionella* contamination, as it can potentially cause disease in the general public and those who are maintaining and replacing these systems. Therefore, this review was focused on understanding the potential risk posed by their increased use and maintenance requirements. This review was approached systematically but was not a full systematic review. There were 61 papers that were considered potentially relevant to the research questions. Of these, 40 papers were considered relevant to the topic of *Legionella* in HWHP and underwent full article assessment and data extraction. The remaining papers were considered useful for background information. The scope of this review established that *Legionella* are a known risk in hot water systems that can be carried over to HWHP systems, yet there is minimal evidence to suggest that the current control measures are being appropriately applied to reduce the risk of exposure. When considering countrywide legislation and guidance, it appears that the risk is considered lower in single- or multi-family homes that do not require a centralised system. This review included the assessment of information regarding the safety of working with HWHP systems with regards to maintenance and replacement. The authors found a lack of information regarding these safety concerns. This review is among the first to systematically evaluate the risks of *Legionella* contamination in HWHP systems.

## 1. Introduction

In Great Britain (GB), the roll out of some 600,000 hot water heat pump (HWHP) systems per year by 2028 has been deemed critical in order to decarbonise residential heating and hot water supplies [1]. The 2021 heating consensus reported that 74% of households in GB use gas-based heating and only a small percentage use other sources, such as electricity [2]. HWHP systems are considered a better alternative, using electricity instead of fossil fuel-based burners and furnaces. Therefore, they will reduce overall energy consumption in line with European Union (EU)-wide targets of full decarbonisation of the building sector by 2050 [3].

The development of HWHP systems is evolving and there are now multiple generations of systems commercially available for single/multiple-family households, district water supplies and commercial buildings. However, there are concerns that these systems are susceptible to microbial contamination because they can hold water at lower temperatures than a standard calorifier. When delivering hot water, these microorganisms may be introduced into facilities, e.g., in households via showers or taps [4]. There is particular concern that microorganisms such as *Legionella* may establish and potentially pose a public health risk to end-users. Notably, *Legionella pneumophila* exposure can cause infection if the bacteria are inhaled, e.g., in water droplets formed during shower use [5]. Legionnaires’ disease can be life-threatening without timely treatment and is a notifiable disease in many countries. Diagnostic/exposure testing has developed in recent years such that antigens can now be tested in urine samples [6]. There are many ways an individual can be exposed to the bacteria and acquire the disease [4]. Examples include exposure from cooling towers, hot tubs, swimming pools, water fountains, etc.

*Legionella* bacteria are generally found in natural water sources, especially in favourable conditions, such as warmer temperatures between 20 °C and 45 °C, stagnant water flow or the presence of free-living protozoa. Protozoa such as amoebae can provide intracellular protection and influence the structure and microbial composition of biofilms. Within a biofilm, *Legionella* can co-exist with other bacterial species that provide essential nutrients and improve conditions. Cyanobacteria and *Flavobacterium breve* have been found to enhance *Legionella pneumophila* growth [7]. An established biofilm can persist and protect *Legionella,* enhancing survival against physical and chemical stressors. In the presence of an established biofilm, *Legionella* can be released continually into the water [8]. *Legionella* can also survive water disinfection procedures [9]. Therefore, they can potentially inhabit potable water in very low numbers. If introduced into a HWHP system, when conditions are appropriate, there is the opportunity for *Legionella* to proliferate.

Despite evidence that *Legionella* are a contaminant found in some hot water systems, it is unclear whether they are likely to pose significant risks to the health of workers and the wider general population if there is wider utilisation of HWHP systems. Furthermore, there is little information with regards to *Legionella* risk to those who work with them. If a HWHP system requires maintenance, replacement of parts or replacement of the whole system, there could be a risk of aerosols being generated in the process and the potential for exposure. It is important to understand the risks posed by the use and maintenance of HWHP systems and, therefore, a rapid evidence review was conducted in order to determine the level of risk with increased instalment of these systems. This review is among the first to systematically evaluate the risks of *Legionella* contamination in HWHP systems.

### Types of HWHP and Risk of Legionella Contamination

HWHP systems can provide buildings with space heating and cooling in addition to the delivery of domestic hot water [10]. They may generate the heat from different sources, such as from air, water or ground. This heat is then amplified by the heat pump system and supplied into the heat distribution system [10]. Air-to-water or ground-to-water HWHP systems need to be able to store water ready for when it is needed. A hot water tank allows the HWHP systems to gradually heat water, with the tank storing it. While most HWHP systems provide water at 55 °C, hot water will periodically need to reach 60 °C to prevent *Legionella* proliferation. Research has shown that almost 90% of *Legionella* bacteria will be inactivated when held at 60 °C [11]. Where water contains 100,000 cfu/mL of *Legionella* or more, the water must be held at 60 °C for 10 min to reduce the bacterial levels below 100 cfu/mL. However, if even small numbers of viable bacteria remain in the system, there is always the chance of recolonisation if the temperature decreases to a more optimal value for the bacteria [12,13].

Some HWHP systems can deliver this temperature; however, most systems also have an electrical immersion heater inside the water storage tank to raise the water temperature periodically when required. Regardless of the heat source, the key concern is that without the correct maintenance and operation of the system, this may provide optimal conditions for the introduction and persistence of microorganisms such as *Legionella*.

The scope of this review encompasses all types of HWHP systems. If *Legionella* are a risk in a HWHP system due to running at lower temperatures that would allow them to flourish, appropriate controls are required to ensure mitigation of *Legionella* growth. For any water heating system, the potential risk of contamination will depend on the following:

The physical design of the system, and whether there is any opportunity for water to remain stagnant in the pipework or a tank for long periods of time, i.e., in the tank or from “dead legs”.

Whether mechanisms are in place to challenge growth such as the use of biocidal treatment, for example, which will prevent or minimise the opportunity for bacteria to grow. The most commonly used mechanism to control *Legionella* growth in such systems is temperature. For example, boosting temperatures regularly, by raising the temperature usually >60 °C for a time-period, which is a technique referred to as thermal disinfection;

Whether the system is used to produce hot water to which individuals are likely to be directly exposed. This includes those who may maintain the system as well as use it for hot water. For example, is the heated water being used from taps and showers, or only being used to feed a closed central heating system?

## 2. Methods

### 2.1. Search Terms and Search Strategy

A rapid evidence review was conducted in relation to *Legionella* and HWHP systems. The review was approached systematically but was not a full systematic review. Search terms specific to the research questions and the scope of the review were determined. Relevant peer-reviewed papers and research reports were identified using search engines and databases, e.g., Web of Science, PubMed and Google Scholar. The search terms were organised in a hierarchy of topics, sub-topics, synonyms and acronyms. Searches were based on the proximity of key terms irrespective of their order based on Boolean operators. The key terms focused on *Legionella* and HWHP systems. The searches were expanded for a variety of HWHP types and included all domestic and commercial set-ups. Terms included control measures for *Legionella* risk as well as any scope of maintenance of the systems. Government guidance and legislation were also added into the search and data extraction to consider what is being circulated in regards to the safety and maintenance of HWHP systems and control of *Legionella* contamination.

### 2.2. Data Extraction and Evidence Summary

To ensure consistency, the evidence from each qualifying study was collated using a data extraction table. An assessment of the design and quality of each study was based on criteria developed for scoping evidence reviews [14]. Evidence was considered in relation to each research question and summarised in terms of its strength, quality and reproducibility. Scores from 1–5 were given for study relevance and applied, 5 being the most relevant, and 1 being the least. Emphasis was made on establishing a link between HWHP systems and potential *Legionella* contamination or exposure.

### 2.3. Relevant Questions

What potentially harmful microorganisms (particularly *Legionella*) have been detected in HWHP systems?

Is there any evidence of respiratory disease because of exposure to microorganisms from HWHP systems?

What control measures are currently in place to prevent exposure to harmful microorganisms from HWHP systems to workers and the public?

## 3. Results

There were 203 papers of potential relevance that were identified through the initial searches. After examining the abstracts and applying inclusion and exclusion criteria, 61 were considered to be potentially relevant to the research questions. Of these, 18 were scored as 5; considered specifically relevant to the topic of *Legionella* in HWHP systems (Summarised in Table 1). A further 22 papers were scored as 4, considered to be relevant to the topic as they made reference to HWHP safety or *Legionella* in household water systems. The remaining papers were considered useful for background information. Those used have been referenced; citations [15,16,17,18,19,20,21,22,23,24,25,26,27,28,29,30,31,32,33,34,35] formed part of the review but not directly referenced [15,16,17,18,19,20,21,22,23,24,25,26,27,28,29,30,31,32,33,34,35].

### 3.1. Presence of Legionella and Risk of Exposure

Many papers acknowledged and considered the requirement of controlling *Legionella* in water heating systems, including those with HWHP systems, but none looked at actual colonisation. Under closer scrutiny, few of these studies experimentally tested for *Legionella*. Most (*n* = 12) of the highest-relevance papers were single experimental studies examining the storage of water. One study by Mathys et al. (2008) tested hot water samples from family residences for the occurrence of *Legionella*. They compared instantaneous hot water heaters to houses that used storage tanks and recirculating hot water [41]. That study showed that there was an increased prevalence (12%) of *Legionella* in homes that were supplied with storage tanks or recirculated hot water. *Legionella pneumophila* accounted for 93.9% of *Legionella* positive samples. A more recent study by Moodley et al. (2023) reported the presence of Acanthamoeba, *Legionella pneumophila*, *Pseudomonas aeruginosa* and non-tuberculous mycobacteria in energy-efficient hot water systems [43]. That study compared samples from winter temperatures (20 °C to 30 °C) to summer temperatures (40 °C to 55 °C). *Legionella pneumophila* were detected in both winter and summer months.

If *Legionella* have been detected in water heating systems, it is reasonable to assume that that there is a potential risk for exposure. However, no research in the scope of this review has demonstrated a direct link between water from HWHP systems and any disease related to *Legionella* directly, and this would be difficult to achieve.

### 3.2. Mitigation of Legionella Growth

The main measure to control the growth of *Legionella* in HWHP systems is to ensure maintenance of temperatures above habitable conditions for the bacteria. The required temperature is dependent on the size and design of the system. There are certain criteria that are generally referred to in order to determine the minimum temperature required to prevent *Legionella* growth. This is reflected in guidance and legislation for protecting health.

According to SIA (2011), the risk of *Legionella* is categorised based on building size and purpose [44]. The temperature of the hot water is considered to be directly related to the presence of *Legionella.* Single-family dwellings and multi-family dwellings without a centralised hot water supply are considered low-risk. Multi-family buildings with a centralised hot water supply are considered medium-risk. Finally larger buildings such as hospitals are considered high-risk. These risk categories are based on whether there is opportunity for heat loss in piping, how much domestic hot water is required, how long the water is stagnant in the system and whether the temperature can be reliably maintained throughout the system via heat tracing. The design of a system must maintain at least 60 °C if there is a storage tank, and circulating water in pipes must maintain ≥55 °C. At the point of use, i.e., at the tap, the water must be ≥50 °C. There are also specifications with regards to maintaining these systems, such as periodical cleaning of tanks for limescale or visible biofilms/build-up. Swiss standards also refer to a thermal shock treatment by increasing the temperature to >70 °C.

France, Austria and Germany refer to the “3 Liter rule”, whereby the minimum required temperature increases if the tubing/piping length delivering hot water to the point of use exceeds 3 L of water. If so, the minimum required temperature should be ≥50 °C for France and Germany, and ≥55 °C for Austrian standards. In larger systems, these standards also refer to the capacity of a storage tank. For example, in France, if the storage tank exceeds 400 L capacity, the temperature must be kept at >55 °C [11]. By the German standard, if the storage tank exceeds 400 L, it must be kept above 60 °C. Annual testing for *Legionella* is also considered a must in both Austria and Germany. According to Austrian Standards Institute (ASI) (2023), if the tested *Legionella* concentration exceeds 100 cfu/mL, restorative action is required [45]. This may include reassessing the system structure, such as replacement of parts, removing “dead legs” or converting to a decentralised domestic hot water system. There are also recommendations for disinfection that include a thermal disinfection at >70 °C. Other options are noted, such as chemical or ultraviolet (UV) disinfection, but thermal disinfection is preferred.

The Japanese technical guidelines from the Ministry of Health, Labour and Welfare identifies measures to reduce the presence of *Legionella* [46]. Typically, water heating is from systems that require tanks that are heated by electricity [11]. Inside the tank, the water must be maintained at 60 °C, and by the time the water reaches the tap it must be 55 °C. There are also stipulations that there must be a “draining valve” present to prevent stagnation and allow for cleaning. In addition, a “flow valve” is a requirement to ensure constant hot water. The tank must be cleaned annually with regular drainage and ensured testing and adjustment of the valves.

In comparison to the reviewed journal documents that refer to thermal inactivation (*n* = 9), the most common temperature range reported to control *Legionella* contamination was between 50 °C and 60 °C. Two documents refer to the use of a “boost”, where the temperature is increased to 60 °C on a daily or weekly basis for the thermal inactivation of *Legionella* bacteria. The minimum and maximum required temperatures of tank storage and circulating hot water vary. Kleefkens (2020) published a thorough review of *Legionella* contamination in HWHP systems, collating this information and reporting a “lack of harmonisation” between countrywide legislation and guidance [11].

### 3.3. Legionella Mitigation and Economic Concerns

Of the reviewed documents, in areas where HWHP systems are used more widely, some papers (*n* = 8) considered the energy efficiency and energy savings of these systems once in place. The higher temperature requirements to reduce the risk of *Legionella* in the system have posed concerns with regards to future energy costs. It was commonly stated that having to increase the temperature to the 60 °C benchmark would increase the overall energy demand and therefore overall costs. Therefore, alternative methods were explored to determine whether this temperature could be reduced while retaining the safety provided by reducing the risk of *Legionella* growth.

Stagnation removal, ensuring that any water storage was continually mixed to maintain the consistent elevated temperature, was highlighted (*n* = 2) as key in reducing the likelihood of *Legionella* contamination. When designing a new building, stagnation and other areas such as piping would be considered in the design. However, in areas such as Great Britain (GB), some of these systems are likely to be retrofitted onto an older, pre-existing system. This could allow for a higher chance of stagnation in pipework or “dead legs”.

Two papers reported that an effective method to reduce the likelihood of *Legionella* being introduced into the system would be to ultrafilter the inlet water. With the addition of ultrafiltration, this also allowed for the system to be kept 5 °C lower, improving energy efficiency [38]. Knapp and Nordel (2017) refer to pasteurising the water prior to introduction with a Duck Foot Heat Exchange model, whereby the model imitates the counter-current heat exchange in the feet of ducks, which preheats the fluid prior to pasteurisation and cools it after. The authors reported minimal *Legionella* contamination two years after installation [39]. *Legionella* can take a considerable amount of time to establish in a system. It requires optimal conditions where a biofilm has formed inside. In order to demonstrate that the probability of contamination has been reduced, it is important to ensure periodic testing for *Legionella* over a longer period of time.

Kaschewski et al. (2020) studied the risk of *Legionella* and the possible methods of energy saving in domestic hot water systems, based on simulation models of a typical single-family home [37]. The results demonstrated that *Legionella* risk was mainly dependent on storage volume and tap water volume. Households that utilised 50 L per day and a 150 L storage tank size remained below 100 cfu/mL of *Legionella*. Simulations based on energy demand for the same system showed that this could be decreased by 62% by operating with “disinfection-on-demand” by boosting the temperature and replacing electric heating of the system with an immersion heater. Furthermore, the authors concluded that integrating a UV Light-Emitting Diode (UV-LED) reduced additional energy demand by 6% and could be an alternative option to thermal disinfection.

None of the reports or publications reviewed expressed or highlighted any concerns relating to the potential for exposure of people who work with or maintain these HWHP systems to *Legionella*. The risks posed to workers from potential exposure when removing a system that has been in place for several years, where it is more likely to have established a biofilm containing *Legionella*, are unknown. This would depend on the potential for aerosolization of water in the system during its removal or replacement of parts. This risk could be mitigated by draining the system and depressurising. Without this evidence, it is difficult to determine whether such work activities may pose a risk in the future.

## 4. Discussion

The review of forty sources demonstrated that *Legionella* are an expected risk of contamination in HWHP systems, particularly where there are issues maintaining temperature or levels of stagnation. Whilst there is an expected risk, little has been done to demonstrate experimentally that *Legionella* contamination is appropriately managed, and this may be difficult to achieve. When testing for *Legionella*, microbiological techniques to test for viable bacteria can be used. However, *Legionella* can be difficult to grow outside of their established biofilm and therefore they can be underrepresented in a given sample. Further molecular techniques such as quantitative Polymerase Chain Reaction (qPCR) can be used to identify and quantify the presence of *Legionella* DNA. However, if the bacterial cells are present but inactivated, this can over-report the levels present in a given sample. Therefore, specific techniques would be required in order to gain a representative quantification of *Legionella* present, which can require specialist testing.

It was difficult to establish evidence of disease directly related to *Legionella* exposure from HWHP systems. There were no sources that made a direct association between exposure from a water source related to a HWHP and Legionnaires’ disease or other *Legionella*-related conditions. However, some sources (*n* = 4) did refer to statistical data that demonstrated an increase in the incidence of Legionnaires’ disease over recent years. In an annual epidemiological report for Legionnaires’ disease, the European Centre for Disease Prevention and Control (2023) reported their highest annual notification rate of Legionnaires’ disease in the European Union, at 2.4 cases per 100,000 population in 2021 [4]. It cannot be ascertained whether the HWHP use has had any impact on this number, as there are other factors that may explain this increase, e.g., an increase in travel after the COVID-19 pandemic. Legionnaires’ disease can also be community-acquired and therefore this increases the difficulty in finding the source of exposure.

In order to establish the link between HWHP systems and Legionnaires’ disease, a water source would need to be positive for *Legionella* and a patient sample would also need to reflect this. Two of the sources from the review (Mathys et al., 2008; Moodley et al., 2023) experimentally analysed water samples for *Legionella*, although some of the relevant review sources expect *Legionella* to be a known contaminant of HWHP systems based on its presence in other hot-water-sourced systems [41,43]. Therefore, whilst there is a lack of evidence, HWHP systems cannot be discounted as a potential source of exposure.

### 4.1. Control Measures

Whilst there is a lack of consistency as to what the most appropriate minimum effective temperature is in controlling *Legionella* growth in HWHP systems, the general consensus is that thermal inactivation is key to controlling system contamination. The most commonly stated method is to maintain the temperature consistently at ≥50 °C, with the option to boost the temperature periodically to 60 °C. However, the recommended boosting temperature also varied between 60 °C and >70 °C. The required time to control this temperature varied from 15 min to 1 h. Emerging evidence suggests that temperature regulation may become problematic in the future due to heat shock resistance, whereby bacteria are able to survive sudden changes in temperature.

A study by Liang et al. (2023) tracked the evolution of heat shock resistance in *Legionella pneumophila* [25]. That study simulated the effects of periodically boosting the temperature against the expression of the heat shock protein present. Temperature treatments were carried out at 50 °C to 59 °C at repeated intervals for approximately 15 min. It was also shown that *Legionella* were found to survive in temperatures of 63 °C. Therefore, to ensure safety, instead of reducing the temperature for economic efficiency, this boosting temperature could require increasing for a longer time frame in the future. This could be as high as 75 °C to 80 °C to allow for bacterial cells to die off faster [11,47]. If small numbers remain in the system, there is the possibility of recolonisation [12,13]. 

Furthermore, if further evidence demonstrates that adaptation to temperature may occur after repeated heat cycles, it may be important to explore other control measures outside of thermal treatment. While these non-thermal control measures are considered less practical or more costly, they are discussed below.

### 4.2. Removal of Bacteria from Water Prior to Introduction into the System

Removing bacteria present in water before it is introduced into the system may be an effective alternative to temperature control measures. The main methods proposed for this were ultrafiltration and pasteurisation of the water. Ultrafiltration physically removes the bacterial cells from the water by allowing only water to pass through, capturing bacteria and other microorganisms on the membrane or barrier that can be removed [48]. The study by Genuardi et al. (2023) demonstrated that with a fitted ultrafiltration device, the hot water supply was continuously found to be free of *Legionella* [38]. In comparison, the cold water was not fitted with a device and *Legionella* were detected. However, ultrafiltration systems require their own maintenance and can be subject to fouling, with pores becoming clogged, reducing their efficiency [49,50]. There are newer technologies for anti-fouling and anti-aging of such membranes [50]; however, this could create further obstacles in terms of costs incurred by introducing additional steps whilst trying to solve the initial issue. Knapp and Nordel (2017) proposed pasteurising the water prior to introduction into the system with a Duck Foot Heat Exchange Model [39]. Whilst this system still uses thermal inactivation, by performing this step prior to introduction into a larger water system, this could effectively prevent *Legionella* contamination taking hold on a larger scale, where exposure could occur. The authors also reported that, with follow-up testing, after two years in circulation there was minimal *Legionella* contamination. It can take a considerable amount of time for *Legionella* to establish in a new system, so it is possible that follow-up tests may demonstrate a slow increase in *Legionella* presence over time. As *Legionella* proliferation requires optimal conditions where a biofilm has formed within the HWHP system, pre-treatment with ultrafiltration or pasteurisation may aid in reducing other bacteria that may generate a biofilm, therefore reducing the potential for *Legionella* to survive.

### 4.3. Ultraviolet (UV) Radiation Treatment

UV treatment has been alluded to in some of the reviewed documents as a possible alternative treatment, but no further analysis was carried out. This treatment method would work by delivering a dose of UV radiation, known to have bactericidal properties, to bacterial cells in the water. UV treatment has been widely researched in water treatment and food preparation [51,52]. However, there are a number of factors that could affect the efficiency of such treatment in HWHP systems, such as ensuring the delivery of the correct dose to efficiently inactivate bacterial cells. Newer technologies are emerging, and UV-LED technology is something that may be explored in more detail in the future. The previously mentioned article by Kaschewski et al. (2020) highlighted that there would be a potential benefit to integrating UV-LED technology in the HWHP system [37]. This was not demonstrated with a clear reduction in *Legionella* cells, but in an additional reduction in energy demand from the system. The authors suggested this technology could potentially replace thermal disinfection in some circumstances.

Conventional UV treatment is delivered from a mercury lamp that emits UV light at 254 nm and has bactericidal effects. Some challenges when using these are that they can be fragile and break, releasing mercury, which poses health and environmental risks. UV-LED is considered a “friendlier” method, as they are a mercury-free option and have also been considered in the application of the treatment of pathogens on food products [48,53]. In addition, they also possess an adjustable wavelength from 210 nm up to visible light, allowing for a tailored delivery of dosage [48]. As with standard UV treatment, there are factors to consider when integrating the technology with a HWHP system; so, whilst promising, there is not enough evidence in the scope of this review to date to suggest this technology is as effective as current methods. However, with increasing interest, more evidence may come to light.

There remains a lack of clarity as to whether these alternative control measures are likely to be effective at reducing the risk of *Legionella* contamination in HWHP systems in the future. This review was particularly interested in the potential risk of exposure to *Legionella* for those who maintain and replace HWHP systems. In the sources reviewed, there was no reference to worker exposure risk during maintenance activities.

## 5. Conclusions

The scope of this review established that *Legionella* presence is an expected risk in HWHP systems, yet there is minimal evidence to suggest that the current control measures are being appropriately applied in order to reduce the risk of exposure. Admittedly, this is difficult to achieve without thorough and targeted testing over prolonged periods of time. There has been an increase in cases of disease related to *Legionella* exposure, but it cannot be ascertained whether this is due to increased instalment of these systems without evidence associating the case(s) of disease with exposure directly from the HWHP system. When considering countrywide legislation and guidance, it appears that the risk is considered lower in single- or multi-family homes that do not require a centralised system. HWHP systems are evolving, and newer generations are being reported to not only reduce *Legionella* contamination but increase efficiency. Whilst thermal inactivation is currently the easiest and most sustainable control method, there are newer technologies that are incorporating potential alternatives. However, little is currently known about their efficacy. This review included the assessment of information regarding the safety of working with HWHP systems in relation to potential exposures to *Legionella* during maintenance and replacement. The authors found a lack of information in this area. Further research is required to address this knowledge gap.

## Figures and Tables

**Table 1 microorganisms-13-01134-t001:** Summary of relevant papers used in this review. The summarised table only includes peer-reviewed articles or reports; other sources from grey literature were discounted.

Reference	Goal of Study	Control Measures	Geographical Location	Future Technology	Type of System Delivery
[36]	Novel concept of non-uniform temperature district heating (NUTDH) system with decentralised heat pumps and standalone heat storage units (HPHS)	Proposed system will increase temperatures in distribution to 70 °C during short periods of time	Central Europe	Yes	District Heating
[37]	Energy savings in hot water supplies and *Legionella* modelling	Removal of stagnation, temperature control and integration of UV-LED technology	Western Europe	Yes	Multi-Family Home/District Heating
[38]	Ultrafiltration plant installation to reduce water temperature and energy demand	Addition of ultrafiltration plant in hot water system	Western Europe	Yes	Multi-Family Home
[39]	Efficiency of Duck Foot Heat Exchange Model	Pasteurisation system	US/Canada	Yes	Domestic Hot Water
[40]	Examination of Energy Refurbishment Packages (ERPs) to lower temperature need	Temperature	Sweden	No	Multi-Family Space Heating/District Hot Water/District Heating
[41]	Testing of hot water systems in randomly selected residences between two cities	Temperature	Germany	No	Single-Family Residences
[42]	Low temperature district heating	No	Western Europe	Yes	District Heating

## Data Availability

No new data were created or analyzed in this study.

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
