# Peer review of "Legionella* in Hot Water Heat Pump (HWHP) Systems"

_microorganisms, 2025, doi:10.3390/microorganisms13051134_

Round 1
Reviewer 1 Report
Comments and Suggestions for Authors
In their ms, the authors surveyed and carefully analysed the published information from reliable sources concerning the presence of Legionella bacteria in heat water heat pump (HWHP) systems and control measures to minimise the risk of exposure. Indeed, the issues (2.3) on harmful bacteria, especially Legionella, in these systems, respiratory diseases to be a result of HWHP contamination, and different measures to reduce the exposure risks are currently topical. Despite the importance of these questions, it would be better to choose another platform to publish this review rather than Microorganisms as much of the information on this bacterium, counting, biofilms, counter-measures is too general for microbiologists and the authors did not focus on new and fascinating findings in microbiology. Why not to publish this work in a journal with more related topics? However, if the Editorial Board of Microorganisms will accept it, OK.
Other comments
Abstract :“ ill health” sounds like an oxymoron
Abstract and elsewhere: As 41 papers were considered relevant, they should all be cited and mentioned in the References section
The section 2 cannot be named as Methods.
The las part in Conclusion as “The authors found a lack of information in this area. Further research is required to address this knowledge gap” is unattractive and adds nothing.
Affiliations and author labels (1-5) should be checked.
Reviewer 2 Report
Comments and Suggestions for Authors
The manuscript presents a review of the potential risks of Legionella contamination in Hot Water Heat Pump (HWHP) systems, particularly in the context of their increasing use in Great Britain by 2028. The review aims to assess the current understanding of Legionella risks in these systems, evaluate existing control measures, and identify gaps in knowledge, especially regarding the safety of maintenance workers and the public.
It is a good review work, however, it relies heavily on secondary data and reviews rather than primary experimental data, which limits its ability to draw definitive conclusions.
Comments and suggestions:
1) Introduction, novelty and objectives
- The introduction provides a good overview of the context and importance of HWHP systems in GB, as well as the potential risks of Legionella contamination.
However, it could be improved by:
i) Expanding on the mechanisms of Legionella proliferation in HWHP systems, including more detailed discussion of biofilm formation and the role of protozoa in protecting Legionella.
ii) Discussing the current state of research on Legionella in HWHP systems, including any previous studies that have directly investigated this issue.
iii) Can the authors provide more details on the potential risks to workers during maintenance and replacement of HWHP systems? Are there any case studies or anecdotal evidence that could support this concern?
- The novelty of the research is not explicitly stated. The review could emphasize that it is one of the first to systematically assess the risks of Legionella in HWHP systems, particularly in the context of their increasing use in GB. A suggested sentence for novelty: "This review is among the first to systematically evaluate the risks of Legionella contamination in HWHP systems, particularly in light of the anticipated widespread adoption of these systems in Great Britain by 2028."
2) Results and Discussion in different sections
- These parts mention the potential for Legionella to adapt to thermal disinfection. Can the authors provide more experimental evidence or case studies to support this claim?
- Authors refer UV treatment and ultrafiltration as potential alternatives to thermal disinfection. Provide more detailed analysis or case studies on the effectiveness of these methods in HWHP systems.
- Section 3.3 mentions the economic concerns of maintaining high temperatures in HWHP systems. Can the authors expand on this, perhaps with a cost-benefit analysis of different control measures?
- Some suggestions of references that could improve the quality of the man manuscript:
i) Bédard, É., Prévost, M., & Détiel, É. (2016). Pseudomonas aeruginosa in premise plumbing of large buildings. Microbiology Open, 5(6), 937–956. This reference could be used in the discussion of biofilm formation and the role of other bacteria in protecting Legionella.
ii) Li, H., Osman, H., Kang, C. W., & Ba, T. (2017). Numerical and experimental investigation of UV disinfection for water treatment. Applied Thermal Engineering, 111, 280–291. This reference could be used in the discussion of UV treatment as an alternative control measure.
iii) Papagianeli, S. D., Aspridou, Z., Didos, S., Chochlakis, D., Psaroulaki, A., & Koutsoumanis, K. (2021). Dynamic modelling of Legionella pneumophila thermal inactivation in water. Water Research, 190, 116743. This reference could be used in the discussion of thermal disinfection and the potential for Legionella to adapt to heat shock.
3) Conclusions
- The conclusion that Legionella is a known risk in HWHP systems is well-supported by the literature reviewed. However, the manuscript speculates on the potential for Legionella to adapt to thermal disinfection over time, citing a study by Liang et al. (2023). While this is a valid concern, the manuscript does not provide sufficient experimental evidence to fully support this claim.
- The authors also speculate on the potential for worker exposure during maintenance, but this is not backed by direct evidence or case studies.
Round 2
Reviewer 2 Report
Comments and Suggestions for Authors
the authors significantly improved the quality of the work in the improved version.